# Osteogenic Differentiation Effect of Human Periodontal Ligament Stem-Cell Initial Cell Density on Autologous Cells and Human Bone Marrow Stromal Cells

**DOI:** 10.3390/ijms24087133

**Published:** 2023-04-12

**Authors:** Jing Wang, Qingchen Qiao, Yaxi Sun, Wenting Yu, Jiran Wang, Minjia Zhu, Kai Yang, Xiaofeng Huang, Yuxing Bai

**Affiliations:** 1Department of Orthodontics, School of Stomatology, Beijing Stomatological Hospital, Capital Medical University, Beijing 100050, China; 2Department of Stomatology, Beijing Friendship Hospital, Capital Medical University, Beijing 100050, China

**Keywords:** cell density, cell proliferation, hPDLSCs, osteoblast differentiation, exosomes

## Abstract

Stem cells have differentiation and regulation functions. Here, we discussed the impact of cell culture density on stem cell proliferation, osteoblastogenesis, and regulation. To discuss the effect of the initial culture density of human periodontal ligament stem cells (hPDLSCs) on the osteogenic differentiation of autologous cells, we found that the hPDLSC proliferation rate decreased with an increase in the initial plating density (0.5–8 × 10^4^ cells/cm^2^) for the 48 h culture cycle. After hPDLSCs induced osteogenic differentiation for 14 days with different initial cell culture densities, the expression of osteoprotegerin (OPG) and runt-related transcription factor 2(RUNX2) and the OPG/ Receptor Activator of Nuclear Factor-κ B Ligand (RANKL) ratio were the highest in the hPDLSCs initially plated at a density of 2 × 10^4^ cells/cm^2^, and the average cell calcium concentration was also the highest. To study hPDLSCs regulating the osteoblastic differentiation of other cells, we used 50 μg/mL of secreted exosomes derived from hPDLSCs cultured using different initial cell densities to induce human bone marrow stromal cell (hBMSC) osteogenesis. After 14 days, the results indicated that the gene expression of OPG, Osteocalcin(OCN,)RUNX2, and osterix and the OPG/RANKL ratio were the highest in the 2 × 10^4^ cells/cm^2^ initial cell density group, and the average calcium concentration was also the highest. This provides a new idea for the clinical application of stem cell osteogenesis.

## 1. Introduction

Mesenchymal stem cells, which are adult stem cells, are undifferentiated cells with self-replication and multidirectional differentiation potential [1]. They have therapeutic applications for substituting or repairing damaged cells through differentiation or secretion effects [2]. In multicellular organisms, cell differentiation does not occur in isolation but by cell-to-cell interactions [3]. This means that cells influence each other’s metabolism and differentiation. Cell–cell interactions affect cell differentiation by affecting cell morphology and adhesion [4].

Bone tissue regeneration is a complex physiological process that is currently a hot topic in clinical research, and a single mechanism cannot explain this phenomenon. Mesenchymal stem cells can secrete a variety of biochemical and immunoregulatory factors to accomplish bone reconstruction [5]. Abundant evidence has indicated that extracellular vesicles containing exosomes, microvesicles, and apoptotic bodies are the mechanism by which stem cells play a therapeutic role in many tissue regenerative processes [6]. Among the mesenchymal stem cells, periodontal ligament stem cells (PDLSCs) are ideal seed cells for applications in osteogenic-oriented research because they have more advantages than other stem cells, such as abundant tissue availability, easy access, and significant tissue regenerative characteristics [7,8]. Recent reports have revealed functional stimuli in stem cells that further improve the immunomodulatory and pro-osteogenic functions of secreted exosomes.

In the process of organ development, disease, repair, and regeneration, the cell density in tissues changes greatly [9]. Studies have shown that the change in cell density has biological significance. Most of the previous experiments focused on changes in the number of cells caused by changes in cell density, but there was little exploration of their biological functions [10]. In an experiment using positive and negative charges to stimulate MC3T3-E1 cells, it was found that there was a difference in cell density near the positive and negative electrodes, and the expression of osteonectin near the negative electrode was significantly increased, the bone trabecula was thicker, and there were more connections between cells, all of which indicated that the difference in cell density had different effects on osteogenesis [11]. Cell density affects the collective organization and movement of epithelial cells by changing the contractility of actin [12]. R. de Grooth reported that in a co-culture of the hemopoietic stem cell line FDCP mix C2GM and fetal rat osteoblasts at a high density (such as 2.5 × 10^4^ cells/cm^2^), it was found that compared with the co-culture of osteoblasts at a low density (such as 1 × 10^4^ cells/cm^2^), the cell alkaline phosphatase activity was higher and the formation of osteoclasts was significantly reduced. Osteoclast formation has been shown to strongly depend on osteoblast density [13]. In an experiment on the influence of cell plating density on the multiplication and osteodifferentiation of umbilical cord stem cells on a calcium phosphate cement (CPC) fibrous scaffold, scholar Hongzhi Zhou concluded that, at lower cell densities, cell vitality and mineralization improved with plating density. Yanhua Wang’s research showed that increasing the plating density of MC3T3 cells promoted the expression of osteogenic genes and the primary mouse bone marrow osteoclast differentiation degree [14]. However, a higher plating density was not necessarily better, and a suitable plating density on CPC produced the best ability of osteodifferentiation and mineralization [15]. Therefore, the above evidence suggests that cell culture density has significant effects on cell behavior, such as migration and differentiation. However, the effects of mesenchymal stem cells on their own osteogenic differentiation at different culture densities and the resulting regulation of osteogenic functions in other cells have rarely been studied.

Exosomes are nano-sized vesicles of various cellular origins [16]. Exosomes can mediate cell-to-cell communication and can be used as cell-free therapeutics [17]. Exosomes that exert their functions through paracrine secretions are the best characterized, and they have been extensively investigated for their ability to deliver bioactive molecules, including proteins, nucleic acids, lipids, and glycoconjugates [18]. Stem cells are a major source of exosome production and are used in different research fields due to their high availability and powerful proliferative potential [19,20]. Whether there is a relationship between exosome function and cell density is unclear. Mesenchymal stem cells with different cell culture densities produce different effects, and whether they are related to exosomes has not yet been studied. The above research needs our further exploration.

In this study, we carried out a series of experiments to investigate the effects of cell seeding density on the proliferative characteristics and osteogenic differentiation of human periodontal ligament stem cells (hPDLSCs) and the effects of exosomes derived from hPDLSCs cultured at different plating densities on the osteogenic differentiation of human bone marrow stromal cells (hBMSCs) through a cell co-culture system.

## 2. Results

### 2.1. hPDLSC Proliferation as a Function of Culture Density

Flow cytometry revealed that hPDLSCs highly expressed mesenchymal stem-cell surface markers (CD146 and CD105) but had very little expression of the hematopoietic stem-cell surface marker CD45 (Figure 1A).

hPDLSCs were inoculated into 6-well plates, and the initial cell density groups were 0.5 × 10^4^, 1 × 10^4^, 2 × 10^4^, 4 × 10^4^, and 8 × 10^4^ cells/cm^2^. After 24 h of culture, there was little contact between cells in the 0.5 × 10^4^/cm^2^ and 1 × 10^4^/cm^2^ groups. This contact and the total cell growth area increased as the initial cell density increased, and the cell groups plated at 1 × 10^4^, 4 × 10^4^, and 8 × 10^4^/cm^2^ covered 50%, 60%, and 80% of the bottom area of the pore plate, respectively (Figure 1B–F,B’–F’,B”–F”). After 48 h of culture, the number of cells, total growth area, and cell connections in each group increased. Cells in the 0.5 × 10^4^, 1 × 10^4^, and 2 × 10^4^/cm^2^ cell density groups were still growing, and the cell–cell connections were limited. In the 4 × 10^4^/cm^2^ and 8 × 10^4^/cm^2^ groups, cells were closely connected, and the percentage of cell connections reached 80% and 95%, respectively (Figure 1G–K,G’–K’,G”–K”).

Trypan blue staining was used to detect cell survival. After 48 h of culture, there was no statistical difference in the survival rate of the hPDLSC cell culture density groups (Figure 1Q; *p* > 0.05), indicating that, with increased cell density, there was no significant cell death. The cell count results after 48 h of culture showed that the proliferation rate decreased significantly as the initial cell culture density increased (Figure 1R, *p* < 0.05). We further measured cell proliferation by PCNA immunocytochemistry staining. It was found that after 48 h of culture, the proportion of PCNA-positive cells in the proliferative state decreased with an increase in the initial plating density, which confirmed that cell proliferation was inhibited with the increase in cell density (Figure 1L–P,S; *p* < 0.05).

### 2.2. Effect of Cell Density on the Expression of OPG, RUNX2, and OPG/RANKL during Osteogenesis of hPDLSCs

To evaluate the effect of cell density on the osteogenic differentiation of hPDLSCs, we induced hPDLSCs using different initial cell densities for 14 days (Figure 2A–E,A’–E’). It was found that the single-cell average calcium concentration in the 2 × 10^4^/cm^2^ initial cell density group was the highest on the fourteenth day (Figure 2F), and the difference was statistically significant (*p* < 0.05). The single-cell average calcium concentration was the total calcium concentration in the well divided by the number of total adherent cells in the well.

The gene expressions of OPG, RUNX2, and RANKL were further determined by RT-PCR. On the fourteenth day of osteogenesis induction, the expression of the OPG (Figure 2G) and RUNX2 (Figure 2H) genes and the OPG/RANKL ratio (Figure 2I,J) in the 2 × 10^4^/cm^2^ initial cell density group were significantly higher than those in the other groups (*p* < 0.05).

### 2.3. hPDLSCs with Different Initial Culture Densities Enable High-Yield Production of Exosomes

To evaluate the production of exosomes by hPDLSCs with different initial culture densities, we cultured hPDLSCs at initial cell densities of 0.5 × 10^4^, 1 × 10^4^, 2 × 10^4^, 4 × 10^4^, and 8 × 10^4^/cm^2^ under the same culture conditions for 3 days. The exosomes were separated from the supernatant by standard methods.

The characteristics of the exosomes were identified, including their morphology, diameter, and typical surface protein expression. The particle size and morphology of the exosomes were observed under transmission electron microscopy (TEM) (Figure 3A–E). The analysis results show that the average particle sizes of the 0.5 × 10^4^, 1 × 10^4^, 2 × 10^4^, 4 × 10^4^, and 8 × 10^4^/cm^2^ initial cell density groups were in the range of exocrine particle sizes (20–500 nm) with a uniform diameter distribution (Figure 3A’–E’,F). The typical cup shape of the exosomes was observed in every group. The Western blotting results showed that, compared with the parent cells, all the exosomes were rich in typical exosomal surface proteins (CD9, CD63, and CD81) but deficient in calnexin (Figure 3G). In general, these results indicate that the exosomes were successfully produced by hPDLSCs.

### 2.4. Osteogenic Effects of hPDLSC Exosomes Derived from Cultures with Different Initial Cell Densities on hBMSCs

To study the effects of different hPDLSC exosomes derived from cultures with different initial cell densities (0.5 × 10^4^, 1 × 10^4^, 2 × 10^4^, 4 × 10^4^, and 8 × 10^4^/cm^2^) on bone regeneration, we used a BCA protein assay kit (Thermo Scientific) to determine the concentration of exosomes according to the manufacturer’s instructions, and we used exosomes at 50 μg/mL to induce the osteogenic differentiation of hBMSCs in each group. First, the exosomes were labeled with Dil red fluorescent dye to ensure the visible internalization of the exosomes in the receptor cells. The fluorescence microscopy results showed that the exosomes in each group entered the cytoplasm of hBMSCs, and the receptor cells had high uptake efficiencies (Figure 4).

Alizarin red staining and quantitative analysis demonstrated that the exosomes of the 2 × 10^4^/cm^2^ initial cell density group significantly enhanced the mineralization of hBMSCs over the 14 days of osteogenesis induction (Figure 5F). The RT-PCR results showed that, compared with other groups, the expression of the osteogenic-related factors osterix (Figure 5G), OCN (Figure 5H), RUNX2 (Figure 5I), OPG (Figure 5J), and OPG/RANKL (Figure 5K,L) was significantly upregulated in the 2 × 10^4^/cm^2^ initial cell density group over the 14 days of osteogenesis induction. In summary, our findings suggest that the 2 × 10^4^/cm^2^ cell culture density group effectively promoted osteogenesis of hBMSCs in vitro.

## 3. Discussion

Previous reports demonstrated that cell morphology and proliferation rates vary depending on the initial cell culture density [21]. By culturing hPDLSCs at different initial cell densities, we found that with an increasing initial cell culture density, the cell survival rates were nearly 100%, but the proliferation rate decreased. As the initial cell culture density increased, the adhesion and colonization area of hPDLSCs gradually increased, the average growth area of a single cell changed from wide to limited, and the intermediate culture density was relatively uniform. We also found that the increases in cell density inhibited cell proliferation, and the failure to cause significant cell death may be due to intercellular contact inhibition, serum starvation, or intracellular hypoxia [22,23].

High cell density/confluency leads to contact inhibition of proliferation (CIP), a fundamental property in which normal cells cease proliferation and cell division when they occupy all the space allocated to them upon reaching confluence. This arrest of cell proliferation is seen in most cells and is associated with a halt in cell division and the initiation of differentiation [24]. Contact inhibition is closely related to the cessation of cell proliferation and is one reason why most cells cultured in vitro produce monolayers of cells [25]. When cells undergo contact inhibition, YAP and TAZ cannot co-transcriptionally regulate the expression of myosin-II genes, leading to the loss of F-actin stress fibers, which damages the formation of the autophagosome. The decreased proliferation of the cells has its origin in contact inhibition and is partially autophagy-dependent, similar to the increased susceptibility to intracellular hypoxia and serum starvation [25]. Guoqin Wang’s research indicated that serum starvation reduced the proliferation of A549 cells [26]. These conclusions are consistent with our experimental results.

In this study, by applying the same concentration of FBS in the same medium during 48 h of culture, serum starvation led to differences in the proliferation rate. In the lower initial cell density groups (0.5 × 10^4^/cm^2^ and 1 × 10^4^/cm^2^), the number of cells almost doubled after 48 h of culture. In the higher initial cell density groups (4 × 10^4^/cm^2^ and 8 × 10^4^/cm^2^), cells did not proliferate as well due to their close connections. Whether there is biological significance to finding an optimum distance between cells without contact inhibition has not been reported yet. Studies on the effects of mesenchymal stem-cell culture density on autogenous osteoblastic differentiation and the osteogenic effects on other cells are also rarely reported. Our important findings confirmed that, with different initial cell densities, hPDLSCs had the strongest autogenous osteoinductive effect in the 2 × 10^4^/cm^2^ initial cell density group and the best osteoinductive effect on hBMSCs in the 2 × 10^4^/cm^2^ initial cell density group.

A previous study reported that the initial cell culture density affected the proliferative activity of human adipose tissue stem cells [27]. This is consistent with our results. Laurie M. Bost’s research showed that changes in the density of human retinal pigment epithelium cells led to differences in the transcription of the basic fibroblast growth factor (bFGF; encoded by FGF2) [28]. However, with an increase in cell culture density, there was no positive correlation trend with intracellular changes. We used hPDLSCs to investigate the effect of cell culture density on self-osteogenesis.

In Ting Liu’s study on the influence of the exosomes of hPDLSCs on the osteogenic differentiation of BMSCs at different osteogenic induction periods, ALP activity and ARS staining showed that exosomes induced by PDLSCS 14 days after osteogenic induction could significantly enhance the osteogenic effect of BMSCs, which was significantly higher than exosomes at other time points. Therefore, we selected 14 days as the time node of osteogenic differentiation induced by hPDLSCs and conducted the following study [29]: After hPDLSCs were seeded at different cell densities, they induced osteogenesis for 14 days. We observed the expression of key osteogenic-related factors (including OPG, RUNX2, and RANKL) and the average calcium ion concentration. OPG belongs to the tumor necrosis factor receptor superfamily and is secreted by osteoblasts. Ref. [30] OPG slows the bone resorption cycle through a strong affinity toward RANKL, which plays an important role in osteoclastic differentiation [31]. There are two forms of OPG, monomeric and dimeric, with the latter present primarily in the extracellular matrix [32]. Osteocalcin (OCN) is a marker of bone formation, specifically secreted primarily by osteoblasts, and is the most abundant calcium-binding non-collagenous protein in bones. Osteoblasts express OCN in the bone matrix during alveolar bone reconstruction [33,34]. OCN plays an important role in bone calcium metabolism and reflects the degree of bone formation and bone transformation. It has significant value in the diagnosis of abnormal calcium metabolism. OCN may play the role of a matrix signal in the recruitment and differentiation of bone-resorbing cells [35,36].

RUNX2, expressed in pre-osteoblasts, is a transcription factor closely related to the osteoblast phenotype. RUNX2 plays an important role in the differentiation of mesenchymal stem cells into osteoprogenitor cells [37]. The RUNX2 signaling pathway influences the expression of osterix, which is the last stage of the osteogenic process [38]. The role of osterix is to induce the differentiation of osteoprogenitors into pre-osteoblasts. The osterix pathway affects the expression of OCN, which indicates that pre-osteoblasts have already differentiated into osteoblasts [39]. The previous report also indicated the differentiation of osteoblasts through the upregulation of RUNX2 and the role of osterix as an inducer. [40] The increase in osteoblasts was followed by an increase in OCN. This mechanism suggested that OCN, together with osteoclasts, played a role in the later stages of bone healing [41].

Osterix is a gene transcription factor identified at the final stage of the differentiation of pre-osteoblast cells into osteoblast cells. Ref. [42] Osterix regulates the final stage of osteogenesis and inhibits chondrogenesis [43]. Osterix is an important downstream factor of the RUNX2 regulation of osteoblasts, which depends on the level of RUNX2 and is only specifically expressed in osteogenic cells [44]. After deposition, some of the MSCs differentiate into osteoblast precursors, and RUNX2 and osterix play an important role in the maturation and redifferentiation of osteoblasts [45].

The OPG/RANK/RANKL signaling pathway is a classic pathway regulating bone reconstruction. RANKL, a member of the tumor necrosis factor superfamily, plays a crucial role in osteoclastic differentiation [46]. RANKL interacts with RANK (nuclear factor receptor activator kappa B, a receptor of RANKL) to activate the downstream signaling molecule nuclear factor κB (NF-κB), which regulates the expression of osteoclast genes [47]. Needless to say, exocrine factors such as Wnt, TGF-b, and Hh can affect the balance of OPG/RANKL/RANK. It is unclear whether these exocrine factors directly or indirectly affect it; further research is needed in future experiments to explore the direct or indirect impact of exocrine factors on OPG/RANKL/RANK.

In this study, we drew the conclusion that as hPDLSC cell culture density increased, cell proliferation decreased. Our innovative discovery was that on the fourteenth day in the process of osteoblastic differentiation of hPDLSCs, the expression of OPG and RUNX2 and the OPG/RANKL ratio peaked in the 2 × 10^4^ cells/cm^2^ initial culture density group, and the average cell calcium concentration was also the highest at this time. These results strongly indicate that modulating hPDLSC cell density can remarkably affect autologous cell bone metabolism and that hPDLSCs with a 2 × 10^4^ initial cell density concentration had an optimum influence on the osteogenic effects.

With a changing initial cell density, the cell proliferation and corresponding cell functions were altered. The question is whether there is an appropriate stem cell density for an optimum osteogenesis therapeutic effect. We further explored the effect of cell culture density on hBMSC osteoblastic metabolism. However, the effect of cell density on the average osteogenic capacity of a single cell has not been reported. Kim implied that cell culture density may be a vital factor for determining the characteristics of MSCs and regulating gene expression patterns [48]. It is important to find the optimal initial cell concentration to determine specifically how stem cells differentiate into other cells.

With bone regeneration becoming a pressing issue in the clinical field, more exosome-related studies on bone remodeling have been conducted. In the bone reconstruction microenvironment, exosomes act on osteoblasts to regulate osteogenesis, promote osteoclast differentiation, and regulate osteoclasis [1]. Hailun Xu showed that BMSC exosomes had osteogenic properties and that titanium-implanted engineered BMSC exosomes promoted osseointegration at the prosthesis-bone interface [49]. Therefore, the modification of the mesenchymal stem cell that secretes the exosomes will affect their osteogenic ability.

It is becoming increasingly important to maintain a cell-type-specific cell density for cell function, but we know very little about how cell density influences cell function in developmental and reparative processes or in response to environmental changes [50]. In the research on the influence of MC3T3 cell density on osteoclastic differentiation of mouse bone marrow cells, the results strongly suggested that regulating cell density significantly affected the resultant bone metabolism, changing the balance between osteoclasts and osteoblasts [14]. In Guangyang Liu’s clinical research on stem cells, the therapeutic effect of the high-dose MSC group on lung injury was better than that of the low-dose MSC group. The conclusion of Kuah demonstrated that the optimal concentration of bone marrow mesenchymal stem cells injected into an articular cavity to treat achondroplasia was 1 × 10^10^/L and cells could not enhance the reparative effect when the concentration was too high [51]. The concentration of stem cells needed for the best therapeutic effect was not the highest. For the first time, we provided new insights into the changes in bone metabolism caused by initial cell density and the resulting changes in the secreted exosomes, explored the molecular mechanism, searched for the best stem cell therapeutic concentration, and provided a theoretical basis for elucidating growth, development, repair, and regeneration.

We carried out a number of experiments. The secreted exosomes of hPDLSCs with different culture densities were extracted to induce the osteogenic differentiation of hBMSCs. We drew the conclusion that the expression of OPG and RUNX2 and the OPG/RANKL ratio were the highest in the 2 × 10^4^ cells/cm^2^ group, and the cell average calcium concentration was also the highest. To summarize, the proliferation rate of hPDLSCs decreased with an increase in the initial culture density. During the process of hPDLSC-induced osteoblastic differentiation, the expression of osterix, OCN, RUNX2, and OPG in the 2 × 10^4^ cells/cm^2^ group reached a peak, and the calcium concentration was also the highest. The exosomes secreted by hPDLSCs in the 2 × 10^4^ cells/cm^2^ group had the strongest osteoinductive effect on hBMSCs. The above results are of great significance for stem cell applications in clinical fields and provide a theoretical basis for future research on bone regeneration by selecting the appropriate stem cell densities and secreted exosomes.

Finally, the specific changes in the contents of the exosomes are not clear. Future research should clearly study the relationship between these factors and the change in the initial cell density. These findings have important practical significance.

## 4. Materials and Methods

### 4.1. Cell Culture and Identification of hPDLSCs and hBMSCs

All experiments on human subjects were conducted according to the Helsinki Declaration. All research was conducted after patients had signed a written informed consent and we had received the approval of the Ethics Committee of the Beijing Stomatological Hospital, Capital Medical University. With the informed consent of patients with an average age of 19 years, we collected freshly extracted orthodontic teeth, scraped 1/3 of the periodontal ligament tissue from the root, digested the tissue with 6 g/L type I collagenase and 8 g/L dispase enzyme, and inoculated the tissue into α-minimal essential medium (MEM) for primary culture of hPDLSCs. Stem cells were identified through flow cytometry. The mesenchymal stem-cell-positive surface markers (CD105 and CD146) and negative surface marker (CD45) were analyzed. The experiment was carried out with 3–5 generations of stem cells.

For the primary culture of hBMSCs, we obtained jaw bone fragments from patients undergoing orthognathic surgery (average age: 28 years) and cut the bone fragments into 1 mm^3^-sized fragments, and the fragments and a single-cell suspension were inoculated in 10% serum α-MEM. We obtained cell colonies about a week later.

### 4.2. Cell Proliferation Test

The hPDLSCs at initial plating densities of 0.5 × 10^4^, 1 × 10^4^, 2 × 10^4^, 4 × 10^4^, and 8 × 10^4^ cells/cm^2^ were inoculated in 6-well plates. After culture for 48 h, the changes in cell density were observed through an inverted microscope (IX73, Olympius, Tokyo, Japan). The nuclei were stained with 2-(4-amidinophenyl)-6-indolecarbamidine dihydrochloride (DAPI, Solarbio, Beijing, China). Specifically, the medium was discarded, and the cells were washed with phosphate-buffered saline (PBS) and fixed with 4% paraformaldehyde. Then, the cells were dyed with 10 μg/mL of DAPI for 5–10 min and observed under a fluorescence microscope. The cell proliferation rate was determined as follows: cell proliferation rate = (number of cells after 48 h of cultur–number of cells initially inoculated)/number of cells initially inoculated.

We applied trypan blue staining to count the number of viable cells. The immunocytochemistry method of proliferating cell nuclear antigen (PCNA) was used to detect the cells in different groups in a proliferating state. Specifically, cells were fixed with 4% paraformaldehyde, permeabilized with 3% Triton X-100 (Sigma, St. Louis, MO, USA), and blocked with normal sheep blocking serum (ZSGB-BIO, Beijing, China) at room temperature for 30 min. Then, the cells were incubated with PCNA antibody (Abcam, Waltham, MA, USA, USA), diluted to a 1:50 ratio, at 4 °C overnight, and incubated with a secondary antibody conjugated to horseradish peroxidase (ZSGB-BIO) for 1 h. The cells were observed under a microscope after DAB staining (ZSGB-BIO). The PCNA-positive cell rate was determined as follows: PCNA-positive cell rate = number of positive cells with deep nuclear staining/total number of cells.

### 4.3. Isolation of Exosomes

After the hPDLSCs were inoculated and cultured with different cell densities, fetal bovine serum (FBS) containing 10% depleted exosomes replaced the α-MEM medium, and cells were cultured for 3 days. The supernatant was then collected for exosome extraction. Extraction was performed by centrifuging at 3000× *g* for 15 min to remove dead cells and debris and then filtering through a 0.22 μm filter (Sigma Aldrich, St. Louis, MI, USA) to remove bacteria and impurities with a diameter greater than 200 nm. The filtrate was filtered with an Ultra15 centrifugal filter unit (100 kDa, UFC910024) (Millipore, Burlington, MA, USA) in accordance with the manufacturer’s instructions. The exosomes were directly used for downstream experiments.

### 4.4. Characteristics of Exosomes

A bicinchoninic acid (BCA) kit was used (Thermo Fisher Scientific, Waltham, MA, USA) to determine the concentration of exosomes. The number and the particle size of exosomes were determined by using a ZetaView PMX 110 (Particle Metrix, London, UK). This was performed by placing 10 μL of exosomes on a copper grid for precipitation and then using 2% phosphotungstic acid for negative staining for 2 min. The morphology of the extracted exosomes was detected and imaged by transmission electron microscopy (HT-7700, Hitachi, Tokyo, Japan).

Western blotting was used to identify the surface markers of hPDLSC exosomes, using the positive markers CD9, CD63, and CD81 and the negative marker calnexin (CST, USA).

3′-Tetramethyllindocarbotyanine perchlorate membrane dye (Dil) (Beyotime, Shanghai, China) was used to label the extracted exosomes. The Dil-labeled exosomes were co-cultured with hBMSCs at 37 °C for 24 h and then observed under a fluorescence microscope (IX73, Olympius, Japan). Fluorescence intensity was quantified using ImageJ software.

### 4.5. Alizarin Red Staining Test

Cells were incubated in osteogenic induction medium (containing 10% FBS, 50 mg/L vitamin C, 10 mM β-sodium glycerophosphate, 10 nM dexamethasone, and α-MEM). After the cells were fixed with 4% paraformaldehyde for 30 min, the procedure followed was strictly in accordance with the manufacturer’s instructions. The alizarin red staining solution (Sigma, USA) was used to stain cells for 10 min at room temperature. An inverted microscope (IX73, Olympius, Japan) was used to observe the calcium-producing nodules. Ten percent cetylpyridinium chloride was used to dissolve the calcium nodules for 30 min. The absorbance at 562 nm was used to measure the concentration of calcium ions.

### 4.6. Reverse-Transcriptase Polymerase Chain Reaction (RT-PCR)

The total RNA of cells was extracted with TRIzol reagent (Ambion, Austin, TX, USA), and the concentration was quantified with a Nanodrop 2000 spectrophotometer (Thermo Fisher Scientific Company, USA). The extracted RNA and a PrimeScript RT kit (Perfect Real Time; TaKaRa, Kusatsu, Japan) were used to synthesize complementary DNA. SYBR Green was used for PCR. The specific gene primer sequences used in this study are shown in Table 1.

### 4.7. Statistical Analysis

All data were analyzed with SPSS 22.0 software. One-way analysis of variance (ANOVA) with an independent sample t-test or Bonferroni correction was used with data that showed a normal distribution. The Wilcoxon–Mann–Whitney test was used to determine the statistical significance of data that were not normally distributed. GraphPad Prism 9 was used for data analysis, and *p* < 0.05 indicated that the difference was statistically significant.

## 5. Conclusions

Cell culture density not only affects the osteogenic induction of stem cells themselves but may also affect the differentiation ability of hBMSCs by secreting exosomes. During the process of inducing the osteoblastic differentiation of hPDLSC autologous cells and hBMSCs, the optimum cell culture density was 2 × 10^4^ cells/cm^2^. This is expected to provide a new laboratory theoretical basis for optimal concentration of stem cell osteogenesis in the clinical application.

## Figures and Tables

**Figure 1 ijms-24-07133-f001:**
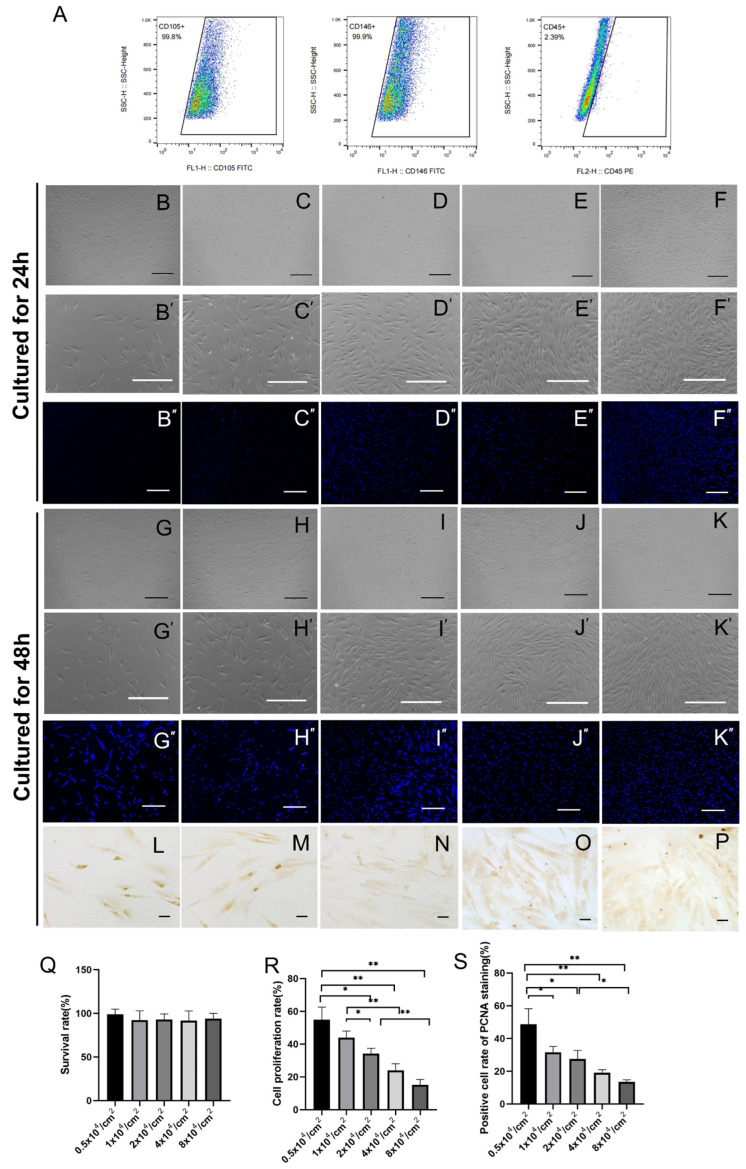
Flow cytometry identification of hPDLSC mesenchymal stem-cell surface markers and cell proliferation characteristics after 24 and 48 h. hPDLSCs showed high expression of mesenchymal stem cell positive surface markers(CD105,CD146), but low expression of hematopoietic stem cell surface markers(CD45) (**A**). hPDLSC morphology was imaged by light microscopy after 24 h and 48 h of culture (**B**–**F**,**B’**–**F’**,**G**–**K**,**G’**–**K’**) and fluorescence microscope imaging of DAPI staining (**B’’**–**F’’**,**G’’**–**K’’**). The cell junctions and total growth area expanded with an increasing initial cell plating density (**B**–**F**,**B’**–**F’**,**G**–**K**,**G’**–**K’**). The hPDLSC survival rate after 48 h of culture was not significantly different according to the cell plating density (**Q**). hPDLSC proliferation significantly declined after 48 h culture as the initial plating density increased (**R**, *p* < 0.05). PCNA immunostaining was used to observe the proportion of proliferating cells (**L**–**P**). After 48 h, the PCNA-positive cell proportion decreased as the initial culture density increased (**S**, *p* < 0.05). * indicates *p* < 0.05, ** indicates *p* < 0.01; Scale bar = 50 μm.

**Figure 2 ijms-24-07133-f002:**
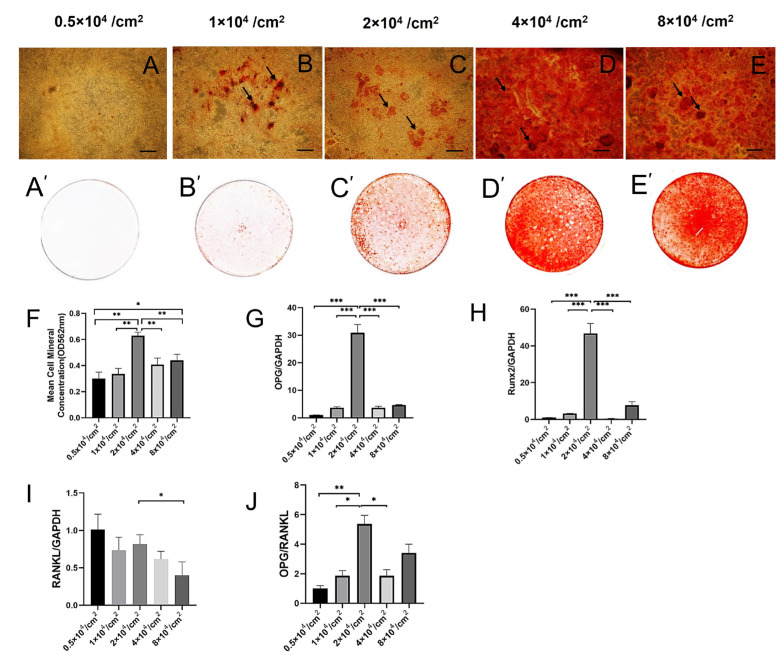
Effect of initial cell density on osteogenesis induction of hPDLSCs over 14 days. Calcium nodules produced through hPDLSC osteogenesis induction were observed under light microscope at different initial culture densities. The black arrows represent calcium nodules. (**A**–**E**) Overview of hPDLSC osteogenesis induction at different initial culture densities. (**A’**–**E’**) Single-cell mean calcium concentration produced by hPDLSC osteogenesis induction in the 2 × 10^4^/cm^2^ cell density group was the highest. (**F**, *p* < 0.05). The expressions of OPG and RUNX2 and the OPG/RANKL ratio were the highest in the 2 × 10^4^/cm^2^ group (**F**–**J**). (*p* < 0.05). * indicates *p* < 0.05, ** indicates *p* < 0.01; *** indicates *p* < 0.001. Scale bar = 50 μm.

**Figure 3 ijms-24-07133-f003:**
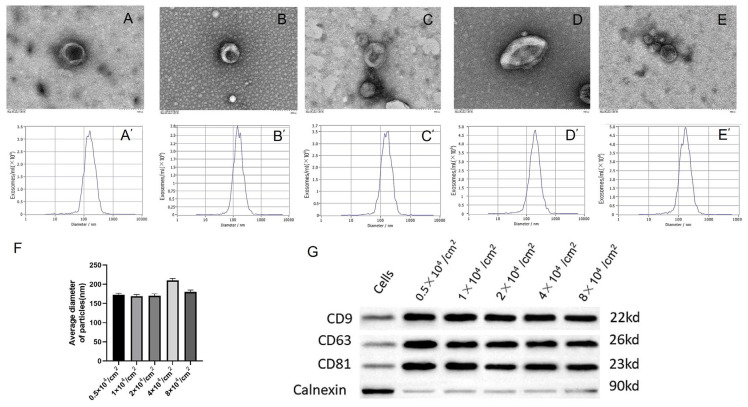
Identification of exosome characteristics from different hPDLSC culture density groups for 3 days of culture (0.5 × 10^4^, 1 × 10^4^, 2 × 10^4^, 4 × 10^4^, and 8 × 10^4^/cm^2^). Representative transmission electron microscopy images of isolated exosomes from different hPDLSC initial cell culture densities (**A**–**E**). Size distribution of exosomes obtained via nanoparticle tracking analysis (**A′**–**E′**). Average particle diameter of isolated exosomes from different hPDLSC initial culture densities (**F**). Western blotting analysis of several representative proteins of isolated exosomes derived from cells (hPDLSCs) plated at 0.5 × 10^4^, 1 × 10^4^, 2 × 10^4^, 4 × 10^4^, and 8 × 10^4^/cm^2^ (**G**).

**Figure 4 ijms-24-07133-f004:**
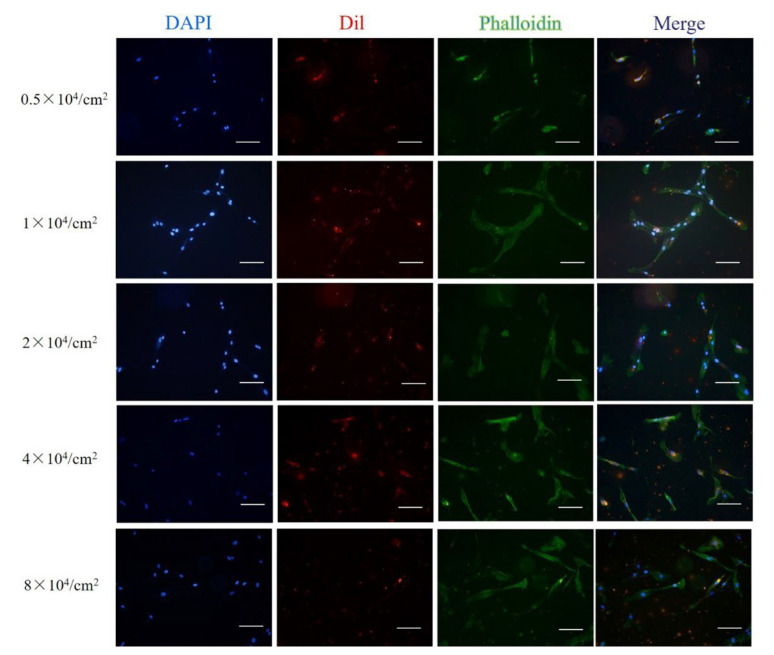
Uptake of secreted exosomes in hBMSCs from hPDLSCs with different initial cell culture densities, detected by fluorescence microscopy. Scale bar: 50 μm.

**Figure 5 ijms-24-07133-f005:**
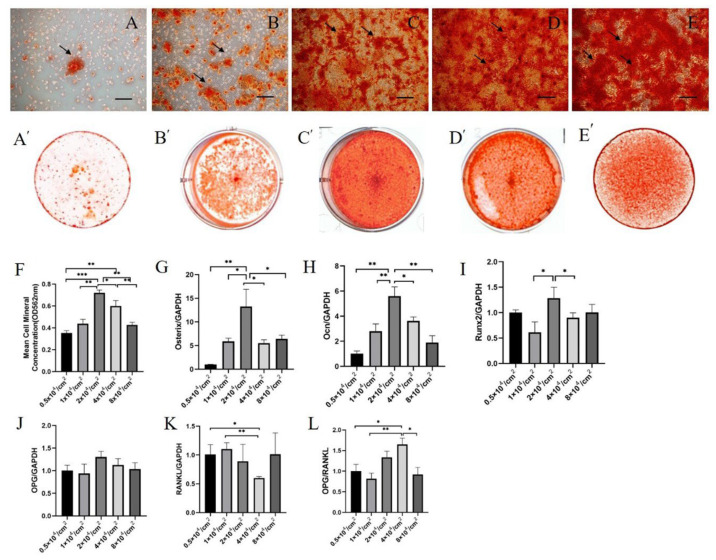
Osteogenesis induction effect over 14 days on hBMSCs by hPDLSC exosomes of different initial cell culture densities. Calcium nodules produced by different hPDLSC culture densities produced exosomes that induced osteogenesis for hBMSCs and were observed under light microscope. The black arrows represent calcium nodules (**A**–**E**). Overview of osteogenesis induction of hBMSCs by hPDLSC culture-derived exosomes from different initial cell culture densities (**A’**–**E’**). The calcium concentration of hBMSC osteogenesis induced by exosomes from the 2 × 10^4^/cm^2^ hPDLSC culture was the highest (**F**). Expressions of osterix, OCN, RUNX2, OPG, and RANKL and the OPG/RANKL ratio in the 2 × 10^4^/cm^2^ group (**G**–**L**). * indicates *p* < 0.05, ** indicates *p* < 0.01, *** indicates *p* < 0.001; scale bar = 50 μm.

**Table 1 ijms-24-07133-t001:** Primer sequences.

Gene (Human)	Forward Primer	Reverse Primer
OPG	5′-GAGTCCGATCCAGCCAAGA-3′	5′-GTACGGCGGAAACTCACAG-3′
OCN	5′-TAGTGAAGAGACCCAGGCGCT-3′	5′-ATAGGCCTCCTGAAAGCCGA-3′
RUNX2	5′-GGAATGCCTCTGCTGTTATGAA-3′	5′-GCTTCTGTCTGTGCCTTCTG-3′
Osterix	5′-CTCCTGCGACTGCCCTAAT-3′	5′-CTCATCCGAACGAGTGAACCT-3′
RANKL	5′-TCGCTGGGAAACAACACTG-3′	5′-GGGAAGGGAAAGGTAGATGC-3′
GAPDH	5′-CTGGGCTACACTGAGCACC-3′	5′-AAGTGGTCGTTGAGGGCAATG-3′

## Data Availability

The data presented in this study are available in the Appendix A.

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
