# Peer review of "Osteogenic Differentiation Effect of Human Periodontal Ligament Stem-Cell Initial Cell Density on Autologous Cells and Human Bone Marrow Stromal Cells"

_ijms, 2023, doi:10.3390/ijms24087133_

Round 1

Reviewer 1 Report

This is an interesting study, and the authors have collected a sufficient dataset using the related methodology. The paper is generally well-written and structured. However, in my opinion, the main aspects of the results and the discussion need to be improved more.

In this study, the authors concluded that as the hPDLSC cell culture density increased, cell proliferation decreased-Line -273. The novelty of the fragment is far from being pointed out, especially given the fact that the are already obtained the same results many years ago. High cell density/confluency leads to contact inhibition of proliferation (CIP), a fundamental property in which normal cells cease proliferation and cell division when they occupy all the space allocated to them upon reaching confluence. This arrest of cell proliferation is seen in most cells and is associated with a halt in cell division and the initiation of differentiation. Contact inhibition is closely related to the cessation of cell proliferation (Timpe et al., 1978) and is one reason why most cells cultured in vitro produce monolayers of cells (Garrod and Steinberg, 1973).

The OPG/RANK/RANKL signaling pathway is a classic pathway regulating bone reconstruction- line268

The OPG/RANKL/RANK system plays an active role in pathological angiogenesis and inflammation as well as cell survival and osteogenesis. At the same time, there are other signal pathways that are more significant for osteogenesis (Wnt, TGF-b, Hh).

I would suggest that the results be presented in a different way. I believe that authors received very interest results - the exosomes secreted by hPDLSCs in the 2 × 104 cells/cm2 group had the strongest osteoinductive effect on hBMSCs. Therefore, cell concentration influences the content of the exosomes produced. This fact can significantly influence cell-free approaches in regenerative medicine.

Reviewer 2 Report

Dear Authors

In your article entitled “Osteogenic differentiation effect of human periodontal liga-2 ment stem cell initial

cell density on autologous cells and human bone marrow stromal cells” your aims were 1)

Identify/Standardize an initial seeding density of hPDLCs which will promote both optimum cell proliferation

rete and mesenchymal cells osteogenic differentiation and 2) consider the osteogenic potential of hPDLCs

derived exosomes on bone marrow mesenchymal cells.

Both topics considered in your research have a high scientific interest, the experimental techniques are wisely

chosen. Although to my opinion it will be better if you will report your research topics in two separate papers

to interpretate extensively your results.

At the present form the article lucks to point out your findings which are presented in a “compact“ manner.

Therefor I suggest, prior publication, to write a more extensive introduction which will include,inform and

justify ALL YOUR EXPERIMENTAL work.

Moreover, I would like to ask some clarifications regarding:

1. The end points. i)Why did you stopped the proliferation assays at 48h, and you have not proceed at

72 hours? II)Do you consider that 14 days of culture are idoneal for mesenchymal cells differentiation

and osteogenesis?

2. Could you please explain why you did not consider the ALP, EGF, Col1a and MMP-2 genes at your

research?

3. Based on which evidences you choose the exosomes concentration?

4. In figure 1, the pictures B’,C’,D’,E’,F’, B’’,C’’,D’’,E’’, and G’,H’,I’,J’,K’, G’’,H’’,I’’,J’’,K’’ seems to have

intense background lighting and noise. I will suggest replacing them.

Round 2

Reviewer 2 Report

no comment